# Venetoclax Decreases the Expression of the Spike Protein through Amino Acids Q493 and S494 in SARS-CoV-2

**DOI:** 10.3390/cells11121924

**Published:** 2022-06-14

**Authors:** Chih-Chieh Chen, Zhi-Jie Zhuang, Chia-Wei Wu, Yi-Ling Tan, Chen-Hsiu Huang, Chia-Yi Hsu, Eing-Mei Tsai, Tsung-Hua Hsieh

**Affiliations:** 1Institute of Medical Science and Technology, National Sun Yat-sen University, Kaohsiung 804201, Taiwan; chieh@imst.nsysu.edu.tw (C.-C.C.); stu995101010@gmail.com (Z.-J.Z.); ten10tan6@gmail.com (Y.-L.T.); 2Rapid Screening Research Center for Toxicology and Biomedicine, National Sun Yat-sen University, Kaohsiung 804201, Taiwan; 3Department of Medical Research, E-Da Hospital/E-Da Cancer Hospital, I-Shou University, Kaohsiung 82445, Taiwan; snoopy79101@gmail.com (C.-W.W.); huangrobert1688@gmail.com (C.-H.H.); 4Department of Obstetrics and Gynecology, Kaohsiung Medical University Hospital, Kaohsiung Medical University, Kaohsiung 80756, Taiwan; husonweihsu@hotmail.com (C.-Y.H.); tsaieing@yahoo.com (E.-M.T.); 5Graduate Institute of Medicine, College of Medicine, Kaohsiung Medical University, Kaohsiung 80756, Taiwan

**Keywords:** SARS-CoV-2, COVID-19, spike protein, virtual screening, molecular docking, molecular dynamics simulations

## Abstract

The new coronavirus disease 2019 (COVID-19) caused by the severe acute respiratory syndrome coronavirus (SARS-CoV-2) has been reported and spread globally. There is an urgent need to take urgent measures to treat and prevent further infection of this virus. Here, we use virtual drug screening to establish pharmacophore groups and analyze the ACE2 binding site of the spike protein with the ZINC drug database and DrugBank database by molecular docking and molecular dynamics simulations. Screening results showed that Venetoclax, a treatment drug for chronic lymphocytic leukemia, has a potential ability to bind to the spike protein of SARS-CoV-2. In addition, our in vitro study found that Venetoclax degraded the expression of the spike protein of SARS-CoV-2 through amino acids Q493 and S494 and blocked the interaction with the ACE2 receptor. Our results suggest that Venetoclax is a candidate for clinical prevention and treatment and deserves further research.

## 1. Introduction

A new type of virus appeared in Wuhan, China, in December 2019 [1,2]. Its potential pathogen was confirmed as a new type of coronavirus. According to phylogenetic evidence, it is closely related to bat coronavirus [3]. By the end of 2019, typical cases of pneumonia had been reported, and the World Health Organization had officially designated it as COVID-19. The International Committee on the Taxonomy of Viruses named the virus SARS-CoV-2 [4,5]. As of 1 December 2021, 270,000,000 COVID-19 cases have been confirmed globally. The top three countries in terms of the number of confirmed cases are the United States, Brazil, and India. There have been more than 5,000,000 COVID-19-related deaths globally (https://covid19.who.int/, accessed on 1 December 2021).

There are currently vaccines and drugs approved for the treatment of COVID-19. However, many therapeutic directions have been actively studied, including ACE2 receptor monoclonal antibody development [6] and convalescent serum therapy [7], TMPRSS2 inhibitors [8], the protease inhibitor drug Paxlovid [9], Lopinavir/Ritonavir [10], and, currently the most discussed, the inhibition of RNA-dependent RNA polymerase via the drugs Molnupiravir [11] and Remdesivir [12]. Among these treatments, we have greater research interest in the ACE2 receptor that can effectively isolate the virus from entering the host cell.

SARS-CoV-2 has a spike (S) glycoprotein that binds to host cell receptors and allows viral nucleic acid sequences to enter the cell through the fusion of the host and the viral membrane [13]. It is currently known that SARS-CoV-2 and Severe Acute Respiratory Syndrome Coronavirus (SARS-CoV) use the same receptor angiotensin-converting enzyme 2 (ACE2) to enter cells [3,8]. Attachment and entry are important for virus propagation, which forces the spike protein to become the main target of vaccines and drugs for the treatment of COVID-19 [14,15].

Drug repurposing by structure-based virtual screening has been highly used in recent years and has been successfully exploited to identify novel drug repositioning opportunities in a wide range of therapeutic areas [16,17,18,19]. Recently, many studies have focused on the repurposing of drugs against SARS-CoV-2 S1-RBD approved by the Food and Drug Administration (FDA) [20,21,22,23,24,25,26]. Senathilake et al. showed that two anthracycline class drugs (Zorubicin and Aclarubicin) and a food dye (E 155) were predicted to be potent inhibitors of S1-RBD and ACE2 interaction [22]. Based on molecular docking, Calligari et al. investigated the structure of S1-RBD and its interactions with antiviral drugs, such as Umifenovir, Pleconaril, and Enfuvirtide [23]. Trezza et al. combined and integrated docking and molecular dynamics simulations to identify the S1-RBD:ACE2 interaction inhibitors Simeprevir and Lumacaftor [24]. Kadioglu et al. applied a workflow of combined virtual drug screening, molecular docking, and supervised machine learning algorithms to identify drug candidates against SARS-CoV-2 [25]. However, currently there are no in vitro studies that provide significant evidence regarding the interactions of these drug candidates with S1-RBD.

## 2. Materials and Methods

### 2.1. High-Throughput Virtual Screening and Molecular Docking

Molecular docking is a widely used technique to generate and score putative protein–ligand complexes based on their calculated binding affinities. In this study, the structure of S1-RBD (PDB ID: 6M0J) was used as the initial coordinates for docking purposes. The binding site for virtual docking was determined by considering the protein residues located ≤8 Å away from the ACE2 binding surface. A total of 1246 FDA-approved small molecules from the ZINC 15 database were downloaded in the SDF format. The SDF file was separated into MOL files according to the record of each compound and then used for virtual screening. Molecular docking was performed using iGEMDOCK v2.1 [27] to generate docked conformations and a ranked lists of drugs according to their docking scores. iGEMDOCK uses a generic evolutionary method and an empirical scoring function consisting of electrostatic, steric, and hydrogen-bonding potentials for molecular docking. The docking parameters were set as follows: the number of docking runs was set to 10 with an initial population size of 300, and the number of generations was defined as 80. Since iGEMDOCK uses an evolutionary approach for conformational sampling in virtual screening, the top ranking of the drug list for each screening is different. To ensure that the most suitable drug was selected, we repeated the virtual screening experiment five times and then selected the drug with the best average ranking as the final target. Subsequently, drugs that were among those with the 20 highest average rankings were then selected for further molecular docking analysis. In molecular docking analysis, the number of solutions was set to 100 for each drug. These 100 docking scores were then used for statistical analysis to evaluate the binding affinity and select the most suitable drug. Finally, the drug with the best average docking score was then applied for further molecular dynamics studies and interaction analysis (Figure 1).

### 2.2. Molecular Dynamics Simulations

The molecular dynamics (MD) simulation was performed using GROMACS version 5.1.4 software [28]. The MD simulations of the complexes S1-RBD:Venetoclax were performed using the CHARMM36 all-atom force field [29]. The Venetoclax topology file was generated by the CHARMM General Force Field server (CGenFF version 4.0 [30]), and the dodecahedron box with periodic boundary conditions was used. In the S1-RBD:Venetoclax simulation, 13,872 TIP3P water molecules were generated, and two of them were replaced by chloride ions to balance the global net charge of +2.00 e. The energy of the model was minimized using the steepest descent algorithm. The simulation was performed in canonical ensembles (NVT) followed by isothermal isobaric ensembles (NPT) with a position-restrained MD simulation for 200 ps, respectively. The MD simulation was performed at a constant pressure and temperature for 50 ns using an integration timestep of 2 fs. The cutoff for nonbonded interactions was 12 Å. The coordinates from the MD simulations were saved every 5000 timesteps. The results were analyzed through various built-in functions from the GROMACS software.

### 2.3. Binding Free Energy Calculation

The binding free energy calculation was performed on the S1-RBD:Venetoclax complex to investigate the protein–drug interactions. The S1-RBD mutants (MTs), Q493Y, S494R, and Q493Y/S494R, were considered in this calculation. The pmx tool was used to construct the hybrid topologies of the system used in the free energy calculations [31,32]. This tool automatically generates hybrid structures and topologies for amino acid mutations that represent the two physical states of the system. Many studies have indicated that the more recent protein force fields (AMBER ff14SB, CHARMM36, and OPLS-AA/M) performed well both in molecular dynamics and free energy perturbation calculations for protein–ligand systems [31,33]. Therefore, the CHARMM36 force field was selected for alchemical free-energy (AEF) calculations, which is thermodynamically consistent with the force field used for ligands. After obtaining this hybrid structure, free energy simulations were performed using GROMACS [28]. From the 10 ns equilibrated trajectories, 100 snapshots were extracted, and a rapid 100 ps simulation was performed starting from each frame. The lambda (λ) ranged from 0 to 1 and from 1 to 0 for the forward and backward integrations, respectively, thus describing the interconversion of the WT-to-MT systems and the MT-to-WT systems, respectively. In this study, to perform a rapid 100 ps simulation, the delta lambda (Δλ, per MD step) was set to 2 × 10^−5^ and −2 × 10^−5^ for the forward and backward integrations, respectively. Therefore 50,000 intermediate lambda values were used to connect the start state with the end state. Finally, the pmx tool was used to integrate the multiple curves and subsequently estimate free energy differences (ΔΔ*G*) using the fast-growth thermodynamic integration approach, which relies on the Bennett Acceptance Ratio (BAR) [34] protocol.

### 2.4. Cell Lines and Drug

Human embryonic kidney (HEK) 293T cell lines were obtained from American Type Culture Collection (ATCC, Manassas, VA, USA). The cells were cultured in Dulbecco’s Modified Eagle’s Medium (DMEM) (Gibco, Grand Island, NY, USA) containing 10% fetal bovine serum (Sigma, St. Louis, MO, USA) and 1% penicillin-streptomycin (Gibco) at 37 °C and 5% CO_2_ atmosphere conditions (Panasonic, MCO-170AICUVDL). The drug of Venetoclax for the cell culture and exposure in the in vitro study was kindly provided by AbbVie.

### 2.5. Transfection and Plasmid

The cells were seeded into a 6-well or 10 cm dish, incubated for 24 h, and were then transfected with plasmid by TurboFect Transfection Reagent (Thermo Scientific, Waltham, MA, USA) according to the manufacturer’s guidelines. The pCMV14-3X-Flag-SARS-CoV-2 S [35] gene plasmids were obtained from Addegne (#145780), and a Q493Y, S494R, and Q493Y/S494R site mutation assay was analyzed through the Protech Corp. gene synthesis lab.

### 2.6. Cell Growth

The cells were seeded into a 96-well, incubated for 24 h, and exposed to dose-dependent Venetoclax for 24 h. The cell growth was analyzed by CCK-8 (Sigma) according to the manufacturer’s guidelines.

### 2.7. Inhibitor Screening Assay

The SARS-CoV-2 Spike-ACE2 interaction was analyzed by the SARS-CoV-2 Neutralization Antibody ELISA Kit (Elabscience, # E-EL-E606) according to the manufacturer’s guidelines. Briefly, Venetoclax (10 and 20 nM) was added to the HRP-RBD working solution, followed by exposure for 4, 8, 16, and 24 h. Afterwards, Venetoclax and the HRP-RBD working solution were added to the well, incubated for 1 h, and were then rinsed three times with a washing buffer. The TMB Substrate Solution was added to each well, and the plate was read at a wavelength of 450 nm for an enzyme-linked immune sorbent assay (ELISA).

### 2.8. Western Blotting

The cells were suspended in a RIPA buffer, and the lysates were stored at −20 °C for a western blotting assay. The cell lysates were resolved by SDA-PAGE, transferred to PVDF membranes, and incubated with primary antibodies, Flag-M2 (#F1804, Sigma) and B-actin (#A2228, Sigma), overnight at 4 °C. The Anti-Flag-M2 antibody was detected in two major bands, 180 kDa and 90 kDa, reflecting a full-length and cleaved spike protein, respectively. The PVDF membranes were analyzed by the Chemiluminescence MultiGel-21 system. Results from three independent experiments were recorded. Western blotting band intensity was quantified by NIH software (ImageJ).

### 2.9. Immunofluorescence Staining

The 2 × 10^4^ cells were seeded on a sterilized cover glass in a 12-well plate. After transfection and exposure to the drug, cells were fixed with 4% paraformaldehydel (Sigma, USA) and blocked with 5% BSA (Sigma). The diluent primary antibodies, ACE-2 (Novus, #171606) and Flag-M2 (#F1804, Sigma), on 293T cells were incubated overnight at 4 °C, followed by Fluor-conjugated secondary antibodies. Cell nuclei were observed in a 4’,6-diamidino-2-phenylindole (DAPI) fixation medium, and the cells were imaged using the NIKON A1 spectral confocal system (Tokyo, Japan). Immunofluorescence intensity was quantified using NIH software (ImageJ).

## 3. Results

### 3.1. Virtual Screening and Molecular Dynamics Simulation of S1-RBD:Venetoclax

In this study, residues present at the interface region of S1-RBD:ACE2 were targeted and used for the structure-based screening and selection of drug molecules using iGEMDOCK v2.1 [27]. The screening was conducted against an FDA-approved subset of the ZINC 15 database and repeated five times (Figure 1). The 20 best-scoring drugs in terms of average ranking were selected, as shown in Appendix A. These drugs were then selected for further molecular docking analysis. The distribution of iGEMDOCK scores of these 20 candidates is shown in Figure 2. The experimental results demonstrate that Venetoclax has the lowest (best) average docking score. The average and best docking scores of Venetoclax are −126.84 and −157.75, respectively. Venetoclax was selected as the best drug candidate for further molecular dynamics studies and interaction analysis. The best docking pose among the top 20 candidates and their interacting residues are shown Figure 3.

Molecular dynamics simulation was performed to understand the stability of our predicted model of the S1-RBD:Venetoclax complex. The trajectory of the S1-RBD:Venetoclax complex was analyzed for the structural (RMSD and radius of gyration (Rg)) and energy (Coul-SR, LJ-SR) properties and for the number of hydrogen bonds as a function of time (Figure 4). The RMSD values of the backbone atoms for the S1-RBD:Venetoclax complex with regard to the initial structure are plotted in Figure 4A. The Rg values for the S1-RBD:Venetoclax complex are shown in Figure 4B. The quality and convergence of the simulated dynamic trajectories can be estimated by assessing the RMSD and Rg values. Based on the simulations of the S1-RBD:Venetoclax complex, the obtained data demonstrated that the trajectories become more stable after 35 ns and with average RMSD and Rg values of 2.71 Å and 18.83 Å, respectively. This phenomenon is also consistent with our interaction energy (Figure 4C) and hydrogen bond (Figure 4D) analysis. The analysis revealed that the trajectory showed a more stable interaction energy and a stable number of hydrogen bonds after 35 ns.

The last conformation of the 50 ns trajectories of the S1-RBD:Venetoclax complex was chosen for the following interaction analysis (Figure 4E). The hydrogen bonds forming residues (Q493 and S494) are highlighted with a ball and stick. The key residues stabilizing the binding sites of S1-RBD:Venetoclax are identified, and the electrostatic potential surface of the S1-RBD:Venetoclax interface is also shown (Figure 4F). We found that the amino acids Q493 and S494 involved in the formation of hydrogen bonds may play an important role in the stabilization of the binding site of S1-RBD:Venetoclax.

### 3.2. Venetoclax Blocked the Interaction between the SARS-CoV-2 Spike Protein and the ACE2 Receptor through Spike Protein Expression Degradation

Next, we used in vitro experiments to verify the important role of Venetoclax in the cell. We used HEK293T cells for drug killing and gene transfer experiments. First, we used CCK-8 to analyze whether Venetoclax has the ability to inhibit the growth of 293T cells. It was found that dose-dependent (5, 10, 20, and 40 nM) Venetoclax did not affect the growth of 293T cells (Figure 5A). We also analyzed a binding assay between the SARS-CoV-2 spike protein and the ACE2 receptor. It was found that the dose-dependent Venetoclax decreased the binding between the SARS-CoV-2 spike protein and the ACE2 receptor at 4, 8, and 16 h. (Figure 5B). Next, we simulated the administration of Venetoclax before exposure (prophylactic group) and after exposure (treatment group) to infection of the COVID-19 virus and detected the expression level of the spike protein in the cell (Figure 5C). We overexpressed the spike protein in the cells to simulate COVID-19 virus infection and found that Venetoclax decreased the level of the spike protein (Figure 5D) in the prophylactic (~0.6 fold) and treatment (~0.59 fold) groups. Previous studies have found that Venetoclax in combination with proteasome inhibitors can improve the effect of treatment [36]. Therefore, we believe that ubiquitin ligase can label endogenous proteins and degrade them. The results show that the proteasome inhibitor MG132 increased degradation of the spike protein when Venetoclax treated the cell (Figure 5E). Finally, we used a confocal fluorescence microscope to verify the protein–protein interaction between the SARS-CoV-2 spike protein and ACE2. Our results found that Venetoclax reduced the interaction between the spike protein and ACE2 in the 293T cell lines (Figure 5F). The immunofluorescence intensity of Venetoclax was low compared to DMSO in the prophylactic (Venetoclax/DMSO = 0.32) and treatment (Venetoclax/DMSO = 0.26) groups. These results suggest that Venetoclax decreased spike protein expression and blocked its interaction with ACE2.

### 3.3. Analysis of the S1-RBD:Venetoclax Complex Structures and Their Binding Free Energy between WT and MT

The simulation results presented in Figure 4E show that the amino acids Q493 and S494 may play an important role in the stabilization of the binding site of S1-RBD:Venetoclax. We compared the sequence variation of the spike protein of various coronaviruses (including SARS CoV-2-Q493Y, SARS CoV-2-S493R, and SARS CoV-2-Q493Y/S493R) at positions 493 and 494 (Figure 6A). The S1-RBD:Venetoclax binding free energy differences between WT and MT structures were calculated in a series of alchemical free energy simulations and using the BAR [34] protocol. The Venetoclax binding affinity differences (ΔΔ*G*) were calculated according to Δ*G*_1_ (Venetoclax-bound)—Δ*G*_2_ (Venetoclax-free). Notably, stabilizing mutations have negative ΔΔ*G* values. We adopted the BAR protocol to calculate the binding affinity differences between S1-RBD and Venetoclax and the point mutations. The resulting binding free energy differences (ΔΔ*G*) between WT and Q493Y-MT, S494R-MT and Q493Y/S494R-MT were 3.34 ± 4.06, −7.86 ± 3.96, and −3.27 ± 5.12 kJ/mol, respectively (Figure 6B–D). The Venetoclax binding affinity was significantly increased in S494R-MT (ΔΔ*G* < 0).

The western blotting results are also consistent with our binding free energy experiments. S493R more strongly decreases expression of the spike protein compared to Q493Y and Q493Y/S493R (Figure 6E–H) in the prophylactic and treatment groups. Based on these results, we believe that SARS CoV-2-S493R increases the activity of the Venetoclax drug, resulting in a decreased level of the SARS-CoV-2 spike protein (Figure 7).

## 4. Discussion

The U.S. Food and Drug Administration (FDA) quickly approved Venetoclax in April 2016 for the treatment of chronic lymphocytic leukemia. Taiwan’s domestic health insurance has also approved Venetoclax for use on patients. Venetoclax is currently the only approved BCL-2 inhibitor that can cause tumor cells to produce apoptosis and achieve the effect of killing tumors [37,38]. Venetoclax binds to plasma protein, and the half-life of the drug is about 14–18 h. The maximum concentration in the blood reaches 5–8 h after taking the drug [39]. Venetoclax is metabolized by liver cell enzymes CYP3A4 and CYP3A5 [40]. Common side effects include nausea, vomiting, diarrhea, neutropenia, fever, and hypokalemia, but the rate of infection is lower than chemotherapy [37].

Recently, based on in silico screening, Venetoclax has been identified as a potential drug candidate against COVID-19. Kadioglu et al. found that Venetoclax bound with high affinity not only to nucleocapsid protein but also to 2′-o-ribose methyltransferase [25]. Chen et al. indicated that Venetoclax bound with high affinity to 3C-like protease [41]. In addition, Venetoclax has been evaluated by virtual screening and MD for binding with the Main protease (Mpro) of SARS-COV-2 [42]. The antiviral activity study also found that Venetoclax showed substantially antiviral activity and enhanced viral death of SARS-CoV-2 [43,44,45,46]. To the best of our knowledge, we are the first to describe Venetoclax activity against the S1-RBD of COVID-19 using both in silico and in vitro approaches through amino acids Q493 and S494. A previous study also found amino acids Q493 located within the van der Waals contact distance between SARS-CoV-2 and human ACE2 [47]. Venetoclax is a closely related orally available analogue of ABT-737, and both of them are BH3 mimetics that can activate multiple pro-autophagic pathways [48]. In addition, Gassen et al. revealed that ABT-737 treatment not only enhances autophagy but also reduces the replication of Middle East respiratory syndrome coronavirus (MERS-CoV) [49]. From these findings and our approaches, it is a reasonable hypothesis that Venetoclax can not only reduce the direct interactions of S1-RBD with ACE2 but also have therapeutic potential by enhancing autophagy.

Currently, few drugs have been approved as effective treatment guidelines for COVID-19, which shows that the medical needs for the development of therapeutic agents have not been sufficiently met. Reducing the entry of viruses into the host is an effective way to slow down the spread and infection of viruses. The FDA has approved the use of drugs in the past to treat new diseases, thus reducing costs and improving efficiency. Remdesivir is one such example, used to develop new research on Ebola virus treatment [50]. Molnupiravir mainly interferes with the genetic code of the virus. When the virus reproduces, the drug that enters the viral RNA will cause mutations to accumulate, making the virus unable to fully replicate and then disappear on its own [11]. Paxlovid belongs to the class of protease inhibitors; its function is to block the enzymes needed for the reproduction of the coronavirus [9]. A previous study also reported that sulfated glycans inhibit SARS-CoV-2 binding with the spike protein. Sulfated glycans and Venetoclax have the same ability to bind to the spike protein [51].

Venetoclax is an FDA-approved clinical drug for the treatment of chronic lymphocytic leukemia (CLC). Therefore, the symptoms of patients who have taken Venetoclax after being infected with COVID-19 are an important reference for future clinical trials. Some case reports indicate that a patient diagnosed with severe COVID-19 pneumonia and concurrent acute myeloid leukemia (AML) had a negative for nasopharyngeal swab test on Day 18 after the administration of Azacytidine-Venetoclax (Week 7 after infection with COVID-19) [52]. A chronic lymphocytic leukemia (CLL) patient on long-term obinutuzumab/Venetoclax tested positive for SARS-CoV-2 RNA after infection with COVID-19, but blood samples were tested for SARS-CoV-2 antibodies. The clinical validation of the weekly antibody test for 6 weeks was all negative [53]. In addition, a large-scale clinical study of CLL (GAIA/CLL13) found that, from December 2016 to September 2019, a total of 926 physically fit and treatment-naive patients in 9 European and Israeli countries were randomized to a standard group and an experimental group (Venetoclax group). Between March 2020 and April 2020, only 7 patients in the GAIA/CLL13 trial were infected with COVID-19, an infection rate of approximately is 1.8% [54]. The above results give us more confidence that Venetoclax can reduce the number of COVID-19 viruses and has therapeutic and preventive effects. Our simulation and basic experiment data found that the clinical drug Venetoclax, which has been approved by the FDA, is a promising drug candidate for the prevention of COVID-19.

## 5. Conclusions

Our in silico and in vitro studies revealed that the drug Venetoclax, approved by the FDA, is a potential therapeutic agent for the treatment of COVID-19 and is worthy of further clinical trials.

## Figures and Tables

**Figure 1 cells-11-01924-f001:**
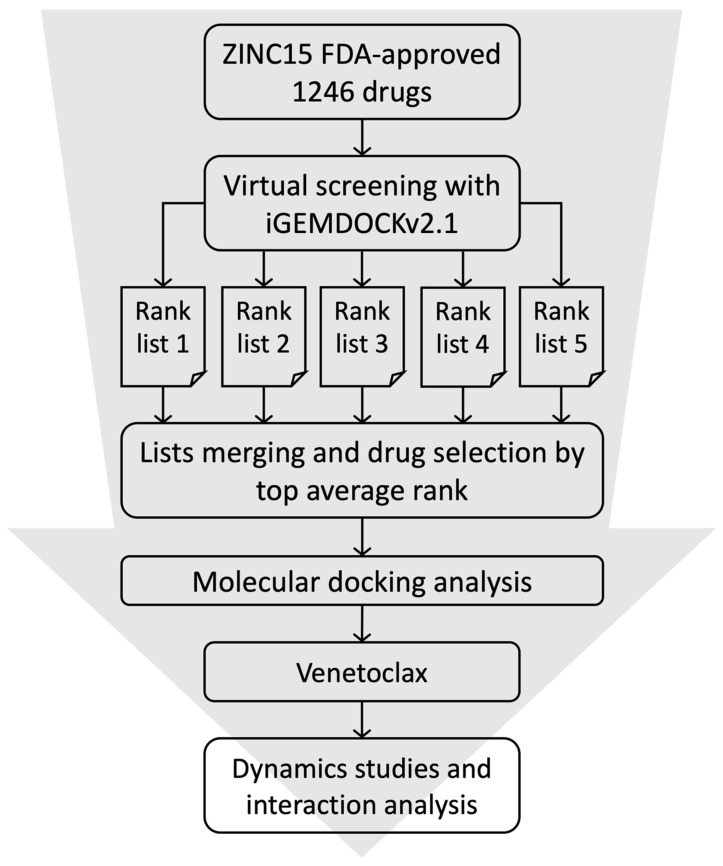
Virtual screening workflow for the identification of the potential S1-RBD binding drugs.

**Figure 2 cells-11-01924-f002:**
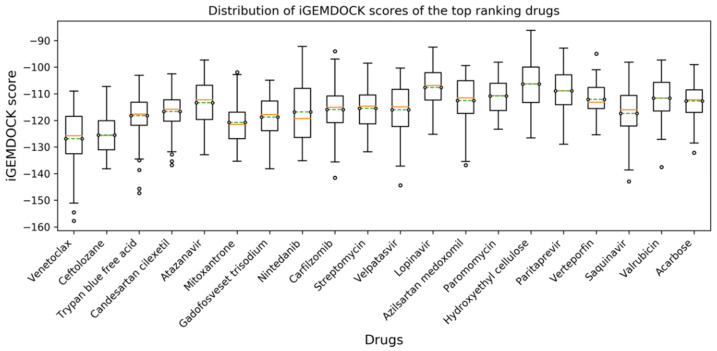
Distribution of iGEMDOCK scores of the top 20 candidates, where a more negative value implies a greater interaction stability.

**Figure 3 cells-11-01924-f003:**
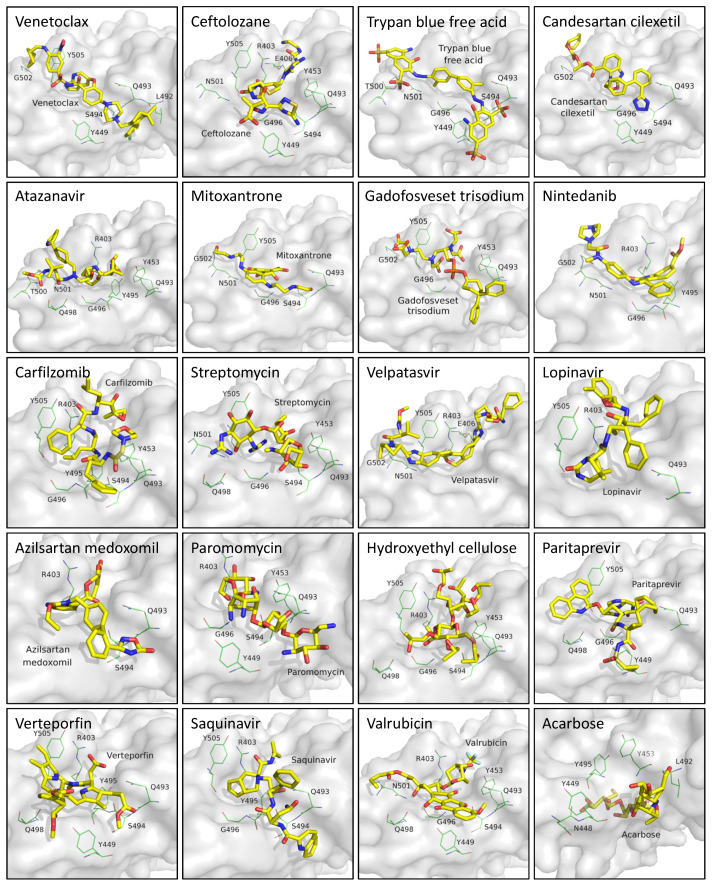
Molecular docking of the top 20 candidates with the best docking scores. The interacting residues are labeled.

**Figure 4 cells-11-01924-f004:**
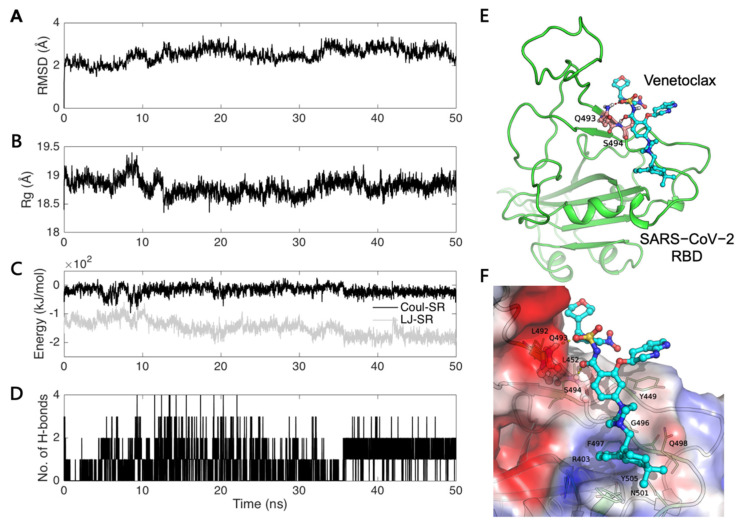
Analysis of the S1-RBD:Venetoclax MD simulation. The trajectory analysis of (**A**) the backbone RMSD and (**B**) the radius of gyration (Rg). (**C**) Coulomb-SR and Lennard-Jones-SR interaction energies, and (**D**) the number of hydrogen bonds of the S1-RBD:Venetoclax interface. (**E**) The last conformation of the 50 ns trajectories of the S1-RBD (green):Venetoclax (cyan) complex. The hydrogen bonds forming residues are highlighted with a ball and stick. (**F**) A close-up view of the S1-RBD:Venetoclax interface. The electrostatic potential surface of the S1-RBD:Venetoclax interface is also shown (blue: positive charge; white: hydrophobic; red: negative charge).

**Figure 5 cells-11-01924-f005:**
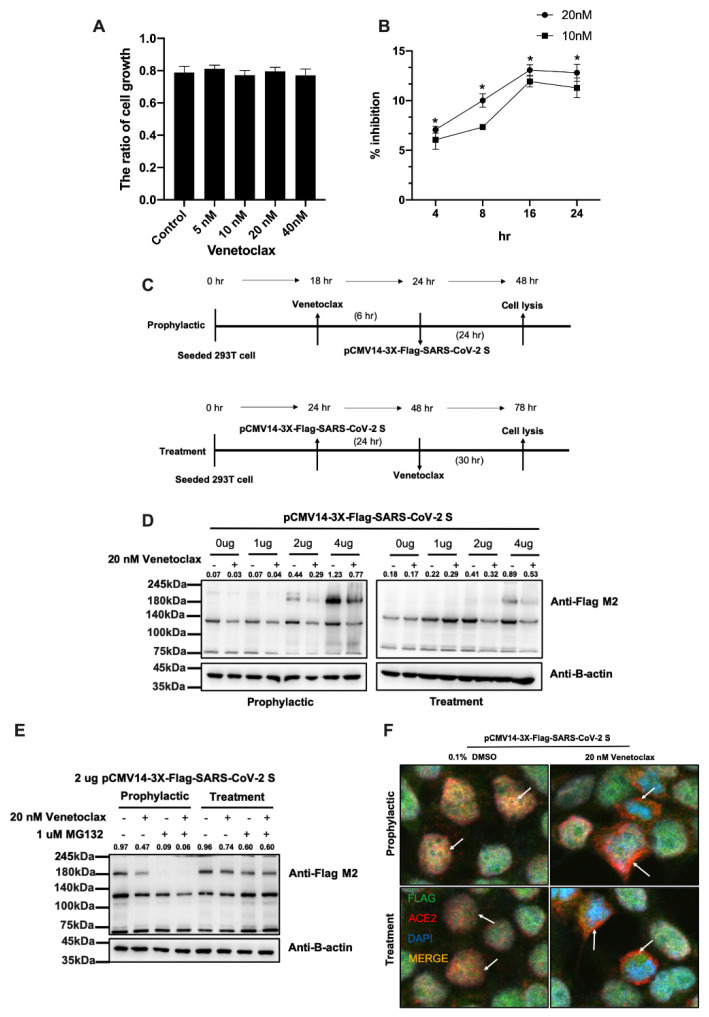
Venetoclax blocked the interaction between the SARS-CoV-2 spike protein and ACE2. (**A**) The cell was treated with 5, 10, 20, and 40 nM Venetoclax, and cell growth was analyzed by a CCK-8 assay at 24 h in HEK293T. (**B**) The inhibitor screening assay of Venetoclax (20 and 40 nM) was analyzed by the SARS-CoV-2 Neutralization Antibody ELISA Kit at 4, 8, 16, and 24 h. (**C**) In the prophylactic group, HEK293T cells were exposed to 20 nM Venetoclax for 6 h and transfected with the SARS-CoV-2 spike protein. After 24 h, cells were collected using a RIPA buffer, and the lysates were stored at −20 °C. In the treatment group, HEK293T cells were transfected with the SARS-CoV-2 spike protein for 24 h and exposed to 20 nM Venetoclax. After 30 h, cells were collected using a RIPA buffer, and the lysates were stored at −20 °C. (**D**) Detection of the flag (180 kDa) by immunoblotting in cells overexpressed with 0. 1, 2, and 4 ug of pcMV14-3X-Flag-SARS-CoV-2 S plasmid. The total level of the flag of the spike protein was detected by immunoblotting. B-Actin was used as a loading control. (**E**) Prophylactic and treatment experiments were performed after adding the proteasome inhibitor MG132 to cells for 1 h. (**F**) The distribution of the flag (Alexa Fluor-488) and ACE2 (Alexa Fluor-555) was determined by immunofluorescence microscopy. Data are the mean values ± SD of three experiments. * *p* < 0.05 vs. untreated control; two-tailed Student’s *t* test.

**Figure 6 cells-11-01924-f006:**
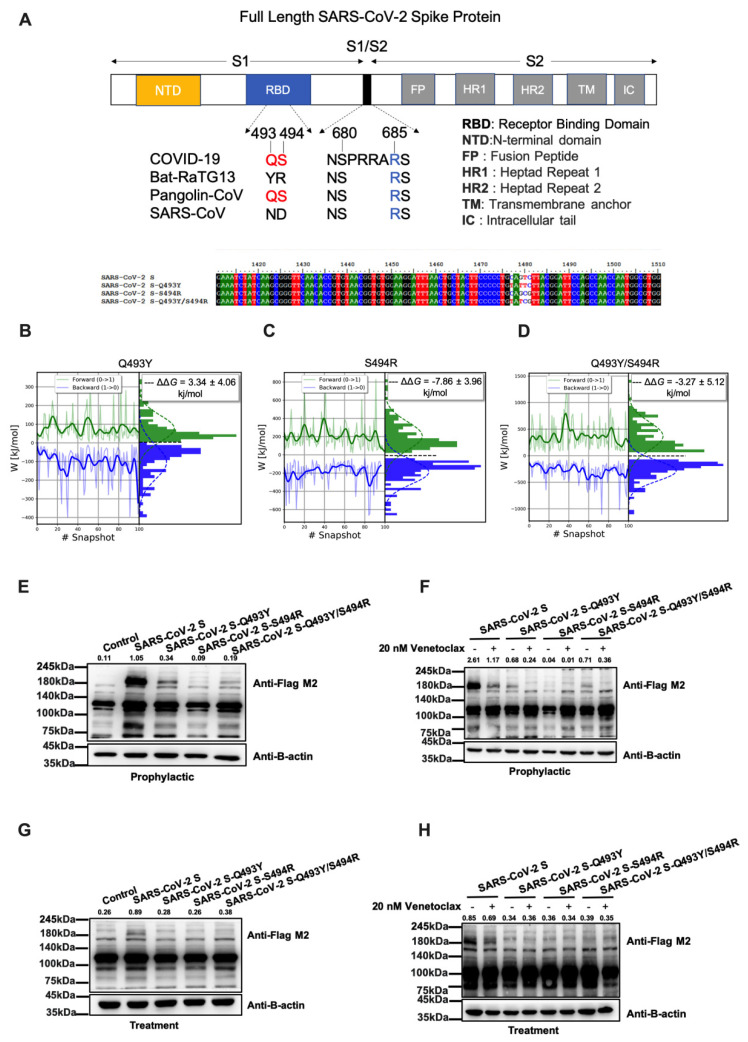
Analysis of the S1-RBD:Venetoclax binding free energy and the level of the SARS-CoV-2 spike protein through S494. (**A**) The sequence variation of various coronaviruses at positions 493 and 494. Distribution of the work values over time for an alchemical free energy simulation of (**B**) Q493Y, (**C**) S494R, and (**D**) Q493Y/S494R mutations. The left plot in each result shows the work values obtained from the integration of δHλ/δλ as a function of the sampling time of the equilibrium states. The right plot in each result shows the histograms of work values from which the free energy was calculated. (**E**–**H**) Detection of the flag (180 kDa) by immunoblotting in cells overexpressed with 2 ug of SARS-CoV-2 S, SARS-CoV-2 S-Q493Y, SARS-CoV-2 S-S494R, and SARS-CoV-2 S-Q493Y/S494R plasmid, respectively. The total level of the flag of the spike protein was detected by immunoblotting. B-Actin was used as a loading control.

**Figure 7 cells-11-01924-f007:**
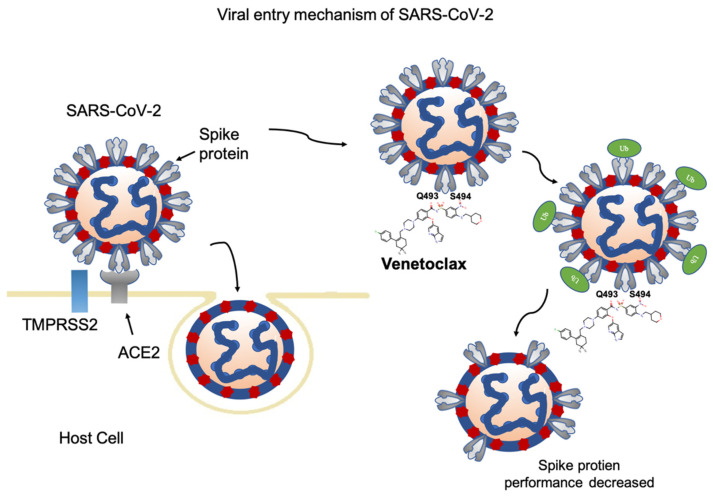
Flowchart showing the methods by which Venetoclax blocks the interaction between SARS-CoV-2 and the ACE2 receptor in the host cell.

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
