# Peer review of "Venetoclax Decreases the Expression of the Spike Protein through Amino Acids Q493 and S494 in SARS-CoV-2"

_cells, 2022, doi:10.3390/cells11121924_

Round 1
Reviewer 1 Report
The manuscript of Dr. Hsieh and co. concerns the effect of Venetoclax, a known drug conventionally used to treat the adults affected by chronic lynfocitic leukemia as a potent binder of the SARS-CoV2 Spike protein. The authors described a rationale of the study that begins from the HTVS of a series of FDA approved-drugs (repurposing) to in vitro phenotypic screenings and binding assays.
The manuscript is interesting and the results sound, but some minor issues have to be addressed.
- The introduction "Drug repurposing by structure-based virtual screening was highly used in the recent 55 years and has been successfully exploited to identify novel drug repositioning opportunities in a wide range of therapeutic areas. Recently, many studies have focused on repurposing of Food and Drug Administration (FDA)-approved drugs against SARS-CoV- 58 2 S1-RBD20-25" should be improved citing recent repurposing studies.
For instance a recent repurposing study: 10.3390/COMPUTATION8030077 - The discussion section should be improved with real comments about the findings of this interesting study. In this form, it seems just an extension of the introduction section
- Line 77 “iGEMDOCK v2.126 to generate the docked conformation of the FDA-approved drugs with a population size of 300 and a number of generations of 80. The number of solutions was set to 10 for each drug…” Please rewrite these sentences
- “The screening was conducted against FDA-approved subset of ZINC 15 database 164 and repeated five times” Could the authors explain better why they performed the docking 5 times?
Author Response
Reviewer 1.
The manuscript of Dr. Hsieh and co. concerns the effect of Venetoclax, a known drug conventionally used to treat the adults affected by chronic lynfocitic leukemia as a potent binder of the SARS-CoV2 Spike protein. The authors described a rationale of the study that begins from the HTVS of a series of FDA approved-drugs (repurposing) to in vitro phenotypic screenings and binding assays.
The manuscript is interesting and the results sound, but some minor issues have to be addressed.
- The introduction "Drug repurposing by structure-based virtual screening was highly used in the recent 55 years and has been successfully exploited to identify novel drug repositioning opportunities in a wide range of therapeutic areas. Recently, many studies have focused on repurposing of Food and Drug Administration (FDA)-approved drugs against SARS-CoV- 58 2 S1-RBD20-25" should be improved citing recent repurposing studies.
For instance a recent repurposing study: 10.3390/COMPUTATION8030077
Response: Following the Reviewer’s comment, we cite this reference in revised manuscript.
- The discussion section should be improved with real comments about the findings of this interesting study. In this form, it seems just an extension of the introduction section
Response: Following the Reviewer’s comment, we added new discussion section in revised manuscript.
- Line 77 “iGEMDOCK v2.126 to generate the docked conformation of the FDA-approved drugs with a population size of 300 and a number of generations of 80. The number of solutions was set to 10 for each drug…” Please rewrite these sentences
Response: Following the Reviewer’s comment, we have rewritten these sentences (Lines 80-85). The main text was adapted as follows. ”The molecular docking was performed using iGEMDOCK v2.1 to generate the docked conformations and ranked lists of drugs according their docking scores. The docking parameters were set as follows: number of docking runs was set to 10 with an initial population size of 300 and the number of generations was defined as 80.”
- “The screening was conducted against FDA-approved subset of ZINC 15 database 164 and repeated five times” Could the authors explain better why they performed the docking 5 times?
Response: Following the Reviewer’s comment, we added the experiment in section 2.1 (Lines 86-89). The main text was adapted as follows.”Since iGEMDOCK uses an evolutionary approach for conformational sampling in virtual screening. Therefore, the top ranking of the drug list for each screening are different. To ensure that the most suitable drug was selected, we repeated five times in the virtual screening experiment and then selected the drug with the best average ranking as the final target.”
Reviewer 2 Report
Authors have shown in vitro data of Venetoclax inhibiting SARS-CoV-2 spike protein binding to ACE2. However, the computational part had been very poorly described to reproduce these results. I have the following comments, which should be addressed before it is accepted for publication:
- The authors should discuss some of the other molecules that have been reported to inhibit SARS-CoV2 binding with spike protein? For example, Sulfated glycans (Dwivedi et al., J Biol Chem. 2021 Oct;297(4):101207) and other?
- Does the iGEMDOCK rank molecule based on their docking score? If so, can the authors provide a histogram of the docking score for all the molecules to see how many of them are binder/nonbinders based on docking alone? It would help understand how crucial this first step of the used workflow is.
- Fig 1D. Are these interaction energies of the ligand with its surrounding? How was that calculated
- Details of the fast-growth thermodynamic integration approach must be mentioned appropriately. Do they have to use a certain number of intermediate lambda values to connect start stated with end-state?
- Is 100 ps MD for each snapshot sufficient enough to get free-energies converged? Can they show that with some plots of gradients or free-energies calculated after every 20ps?
- The resulting binding free energy differences (ΔΔG) between WT- and Q493Y-, S494R- and Q493Y/S494R-MT were 3.34 ± 4.06, -7.86 ± 3.96 and 3.19 ± 3.66 kJ/mol. What happens in the case of double mutant Q493Y/S494R that the resultant free-energies are not the some of Q493Y and S494R mutant?
- How do these errors in free-energy were calculated?
- The accuracy of the AFE calculations may vary with the choice of the force field (J. Chem. Theory Comput. 2015, 11, 7, 3333–3345). Can the author discuss that in the paper and mention what made them choose this force-filed?
- In half of the places, "spike protein" is written as "spick protein". Authors should check the manuscript carefully.
Author Response
Reviewer 2.
Authors have shown in vitro data of Venetoclax inhibiting SARS-CoV-2 spike protein binding to ACE2. However, the computational part had been very poorly described to reproduce these results. I have the following comments, which should be addressed before it is accepted for publication:
1. The authors should discuss some of the other molecules that have been reported to inhibit SARS-CoV2 binding with spike protein? For example, Sulfated glycans (Dwivedi et al., J Biol Chem. 2021 Oct;297(4):101207) and other?
Response: Following the Reviewer’s comment, we added new discussion section in revision manuscript. “Previous study also reported that Sulfated glycans inhibit SARS-CoV-2 binding with spike protein. Sulfated glycans and Venetoclax have the same ability to bind to Spike protein [46]”
2. Does the iGEMDOCK rank molecule based on their docking score? If so, can the authors provide a histogram of the docking score for all the molecules to see how many of them are binder/nonbinders based on docking alone? It would help understand how crucial this first step of the used workflow is.
Response: Yes, iGEMDOCK rank molecule based on their docking score. As per review's suggestion, we conducted a further docking analysis for the top 20 drugs with the best average ranking. In molecular docking analysis, the number of solutions was set to 100 for tach drug and these 100 docking scores were then used for statistical analysis to evaluate the binding affinity and select the most suitable drug. Figure 2 and Figure R1 display the distribution of the docking scores for the top 20 drugs. These experimental results demonstrate that Venetoclax has the lowest (best) average docking score compared with others. The average and the best docking scores of Venetoclax are -126.84, and -157.75, respectively.
Result in the PDF file
Figure R1. Distribution of iGEMDOCK scores of the top 20 candidates, where a more negative value implying greater interaction stability.
3. Fig 1D. Are these interaction energies of the ligand with its surrounding? How was that calculated
Response: Following the Reviewer’s comment, to quantify the strength of the interaction between S1-RBD and Venetoclax, the short-range nonbonded interaction energies (Coulomb-SR and Lennard-Jones-SR) between S1-RBD and Venetoclax were decomposed by GROMACS package.
4. Details of the fast-growth thermodynamic integration approach must be mentioned appropriately. Do they have to use a certain number of intermediate lambda values to connect start stated with end-state?
Response: Following the Reviewer’s comment, we added the experiment in section 2.3 (Lines 131-134). The main text was adapted as follows. The fast-growth thermodynamic integration (TI) approach relies on Jarzynski's equality [1] (when transition is performed in one direction only) and on the Crooks Fluctuation Theorem (CFT) [2] (when the transitions are performed in both directions). In the fast-growth method, the free energy difference in the folded state of a protein is calculated from two independent equilibrium simulations, WT and mutant. These simulations need to sufficiently sample the end state ensembles, as the free energy accuracy will depend on the sampling convergence. In this study, for a rapid 100-ps simulation the delta lambda (Δλ, per MD step) were set to 2e-5 and -2e-5 for the forward and backward integrations, respectively. Therefore 50000 intermediate lambda values were used to connect start stated with end-state.
5. Is 100 ps MD for each snapshot sufficient enough to get free-energies converged? Can they show that with some plots of gradients or free-energies calculated after every 20ps?
Response: Thanks for the reviewer’s suggestions. Using Q493Y as an example, free-energies were calculated based on 20 ps, 50 ps, and 100 ps MD for each snapshot and the free-energies were 3.53 ± 3.06 kJ/mol, 3.63 ± 23.96 kJ/mol and 3.34 ± 4.06 kJ/mol, respectively (Figure R2). The results show that 100 ps MD simulation is enough for free-energies calculation in this study.
Result in the PDF file
Figure R2. Distribution of the work values over time for 20 ps, 50 ps and 100 ps alchemical free energy simulation of the Q493Y mutation.
6. The resulting binding free energy differences (ΔΔG) between WT- and Q493Y-, S494R- and Q493Y/S494R-MT were 3.34 ± 4.06, -7.86 ± 3.96 and 3.19 ± 3.66 kJ/mol. What happens in the case of double mutant Q493Y/S494R that the resultant free-energies are not the some of Q493Y and S494R mutant?
Response: We thank the reviewer for pointing out this issue. We re-calculated the binding free energy differences in the case of double mutant Q493Y/S494R. The new ΔΔG is equal to -3.27 ± 5.12 kJ/mol (Figure R3), which is more reasonable and consistent with our experimental results.
Result in the PDF file
Figure R3. Distribution of the work values over time for an alchemical free energy simulation of Q493Y/S494R mutations. The left plot in each result shows the work values obtained from the integration of δHλ/δλ as a function of the sampling time of the equilibrium states. The right plot in each result shows the histograms of work values from which the free energy was calculated.
7. How do these errors in free-energy were calculated?
Response: Following the Reviewer’s comment, the free-energy analyses were carried out using the pmx tool. Uncertainties were obtained via bootstrap. Where multiple independent calculation repeats were performed, the standard error for a ΔG estimate was calculated by incorporating both the estimator uncertainty as well as the variance of the repeated calculations. For each repeat, the distribution of ΔG values was assumed to be normal, with mean equal to ΔG estimate and standard deviation equal to the bootstrapped uncertainty. The final uncertainty estimate considering all repeats was calculated as a standard error across all the normal distributions.
8. The accuracy of the AFE calculations may vary with the choice of the force field (J. Chem. Theory Comput. 2015, 11, 7, 3333–3345). Can the author discuss that in the paper and mention what made them choose this force-filed?
Response: Following the Reviewer’s comment, we added the experiment in section 2.3 (Lines 121-126). The main text was adapted as follows. Many studies have indicated that the more recent protein force fields (AMBER ff14SB, CHARMM36, and OPLS-AA/M) performed well both in molecular dynamics and free energy perturbation calculations for protein-ligand systems [3,4]. Therefore, the CHARMM36 force field was selected for alchemical free-energy (AEF) calculations, which is thermodynamically consistent with the force field used for ligand.
9. In half of the places, "spike protein" is written as "spick protein". Authors should check the manuscript carefully.
Response: Following the Reviewer’s comment, we have made the correction accordingly in the revision manuscript.

Reviewer 3 Report
The authors performed a virtual screening study, using ZINC 15and Drugbank databases, to test several drugs as inhibitors of the interaction between the spike (S) glycoprotein of the severe acute respiratory 18 syndrome coronavirus (SARS-CoV-2) and the human receptor angiotensin converting enzyme 2 (ACE2). FDA-approved drugs were tested by means of molecular docking (MD) with the S1 receptor binding domain (RBD) of the spike S protein as receptor. The best average ranking value was used to determine the performance obtained with each drug molecule. Through this test, twenty drug candidates were proposed, being Venetoclax drug the one with the top-score amongst the candidates. Thus, Venetoclax was selected to perform several molecular dynamics simulations to elucidate the ligand-protein interactions and to estimate the free energy differences at standard conditions. In addition to the in silico studies, in vitro evaluations were performed as well. Human embryonic kidney cells were treated with Venetoclax and transfected with gene plasmid containing the information of SARS-CoV-2 and to subsequently analyze the interaction with the ACE2 receptor with the spike S. Inhibition of the binding between spike S protein and the ACE2 rceptor was evaluated as well. Finally, distribution of flag, Alexa Fluor-488, and ACE2 (Alexa Fluor-555) was determined by seberal immunofluorescence microscopies. Authors concluded that the FDA-approved Venetoclax drug may be a potential agent for the treatment of the disease produced by the SARS-CoV-2 (COVID-19) as well as propose further research in medical trials.
- Please, include supporting information about the posses with the highest score, for the twenty candidates, to provide reproducible information for other docking studies on SARS-COV-2. A table with the interacting residues or figures with the structure around the binding sites are suggested.
- Since iGEMDOCK is not a widely used software for docking such as Vina, AutoDock 4, GLIDE or GOLD, please include a brief description of the software and theoretical method used.
- Also, there is a relationship between score and ligand-protein binding energy? The above since positive values of binding energy are related to energetically unstable conformations.
- How did you prepare the molecules prior to docking? What optimization procedure was followed? Please, add the necessary details in the methods subsection.
- Please, include a brief introduction to the docking studies on inhibition of SARS-COV-2/ACE2.
- Although this work is believed to be the first report of that suggests the use of Venetoclax for COVID-19 treatment, there are some references linking the disease with chemotherapy containing this drug. Please, read the following references and extract a few https://dx.doi.org/10.4084%2FMJHID.2021.057, https://doi.org/10.1093/jalm/jfaa137, https://doi.org/10.1038/s41375-020-0941-7,
- In addition, Venetoclax has been evaluated by virtual screening and MD for binding with the main protease MPro of SARS-COV-2. https://doi.org/10.1080/07391102.2021.1967786. Please, include this reference pointing to the therapeutical potential of Venetoclax.
- Please include a discussion why Kd measurements are not included in this study? and justification why they are not necessary
- Please, reformulate the conclusions subsection to highlight the most important points achieved by your analyses. Quantitative values must support your conclusions.
Author Response
Reviewer 3.
The authors performed a virtual screening study, using ZINC 15and Drugbank databases, to test several drugs as inhibitors of the interaction between the spike (S) glycoprotein of the severe acute respiratory 18 syndrome coronavirus (SARS-CoV-2) and the human receptor angiotensin converting enzyme 2 (ACE2). FDA-approved drugs were tested by means of molecular docking (MD) with the S1 receptor binding domain (RBD) of the spike S protein as receptor. The best average ranking value was used to determine the performance obtained with each drug molecule. Through this test, twenty drug candidates were proposed, being Venetoclax drug the one with the top-score amongst the candidates. Thus, Venetoclax was selected to perform several molecular dynamics simulations to elucidate the ligand-protein interactions and to estimate the free energy differences at standard conditions. In addition to the in silico studies, in vitro evaluations were performed as well. Human embryonic kidney cells were treated with Venetoclax and transfected with gene plasmid containing the information of SARS-CoV-2 and to subsequently analyze the interaction with the ACE2 receptor with the spike S. Inhibition of the binding between spike S protein and the ACE2 rceptor was evaluated as well. Finally, distribution of flag, Alexa Fluor-488, and ACE2 (Alexa Fluor-555) was determined by seberal immunofluorescence microscopies. Authors concluded that the FDA-approved Venetoclax drug may be a potential agent for the treatment of the disease produced by the SARS-CoV-2 (COVID-19) as well as propose further research in medical trials.
1. Please, include supporting information about the posses with the highest score, for the twenty candidates, to provide reproducible information for other docking studies on SARS-COV-2. A table with the interacting residues or figures with the structure around the binding sites are suggested.
Response: As per review's suggestion, we have provided the interacting residues around the binding sites in Figure R4.
Result in the PDF file
Figure R4. Molecular docking of the top 20 candidates (yellow) with the best docking score. The interacting residues (green) are labelled.
2. Since iGEMDOCK is not a widely used software for docking such as Vina, AutoDock 4, GLIDE or GOLD, please include a brief description of the software and theoretical method used.
Response: As per review's suggestion, we added the experiment in section 2.1 (Lines 82-84). The main text was adapted as follows. iGEMDOCK uses a generic evolutionary method and an empirical scoring function consists of electrostatic, steric, and hydrogen-bonding potentials for molecular docking.
3. Also, there is a relationship between score and ligand-protein binding energy? The above since positive values of binding energy are related to energetically unstable conformations.
Response: Following the Reviewer’s comment, iGEMDOCK uses an empirical scoring function and an evolutionary approach for flexible ligand docking. The program provides the energy-based scoring functions for all screening compounds which consists of electrostatic, steric, and hydrogen-bonding potentials. Figure R1 shows the distribution of iGEMDOCK scores, where a more negative value implying greater interaction stability.
4. How did you prepare the molecules prior to docking? What optimization procedure was followed? Please, add the necessary details in the methods subsection.
Response: As per review's suggestion, we added the experiment in section 2.1 (Lines 77-80). The main text was adapted as follows. A total of 1,246 FDA-approved small molecules from ZINC 15 database were downloaded in the SDF format. The SDF file was separated into MOL files according to the record of each compound and then used for virtual screening.
5. Please, include a brief introduction to the docking studies on inhibition of SARS-COV-2/ACE2.
Response: Following the Reviewer’s comment, we added a brief introduction to the docking studies in the introduction section (Lines 56-70). The main text was adapted as follows. Drug repurposing by structure-based virtual screening was highly used in the recent years and has been successfully exploited to identify novel drug repositioning opportunities in a wide range of therapeutic areas [5-8]. Recently, many studies have focused on repurposing of Food and Drug Administration (FDA)-approved drugs against SARS-CoV-2 S1-RBD [9-15]. Senathilake et al. showed that two anthracycline class drugs (Zorubicin and Aclarubicin) and a food dye (E 155) were predicted to be potent inhibitors of S1-RBD and ACE2 interaction[11]. Based on the molecular docking Calligari et al. investigated the structure of S1-RBD and its interactions with antiviral drugs, such as Umifenovir, Pleconaril and Enfuvirtide[12]. Trezza et al. combined and integrated docking and molecular dynamics simulations to identify the S1-RBD:ACE2 interaction inhibitors, Simeprevir and Lumacaftor [13]. Kadioglu et al.applied a workflow of combined virtual drug screening, molecular docking and supervised machine learning algorithms to identify drug candidates against SARS-CoV-2 [14]. However, currently no in vitro studies have provided significant evidence regarding the interactions of these drug candidates with S1-RBD.
6. Although this work is believed to be the first report of that suggests the use of Venetoclax for COVID-19 treatment, there are some references linking the disease with chemotherapy containing this drug. Please, read the following references and extract a few https://dx.doi.org/10.4084%2FMJHID.2021.057, https://doi.org/10.1093/jalm/jfaa137, https://doi.org/10.1038/s41375-020-0941-7,
Response: Following the Reviewer’s comment, we added new discussion section in revised manuscript. “Venetoclax is an FDA-approved clinical drug for the treatment of chronic lymphocytic leukemia (CLC). Therefore, the symptoms of patients who have taken Venetoclax after being infected with COVID-19 are an important reference for future clinical trials. Some case reports indicate that a patient diagnosed with severe COVID-19 pneumonia and concurrent acute myeloid leukemia (AML) had a negative for nasopharyngeal swab test on day 18 after administration of Azacytidine-Venetoclax (week 7 after infection with COVID-19) [47]. Chronic lymphocytic leukemia (CLL) patient on long-term obinutuzumab/venetoclax tested positive for SARS-CoV-2 RNA after infection with COVID-19, but blood samples were tested for SARS-CoV-2 antibodies. The clinical validation of the weekly antibody test for 6 weeks was all negative[48]. In addition, a large-scale clinical study of CLL (GAIA/CLL13) found that from December 2016 to September 2019, a total of 926 physically fit and treatment-naive patients in 9 European and Israeli countries were randomized to standard group and experimental group (Venetoclax group). Between March 2020 and April 2020, only 7 patients in the GAIA/CLL13 trial were infected with COVID-19, an infection rate of approximately is 1.8%[49]. The above results give us more confidence that venetoclax can reduce the number of COVID-19 viruses and has therapeutic and preventive effects.”
7. In addition, Venetoclax has been evaluated by virtual screening and MD for binding with the main protease MPro of SARS-COV-2. https://doi.org/10.1080/07391102.2021.1967786. Please, include this reference pointing to the therapeutical potential of Venetoclax.
Response: Following the Reviewer’s comment, we cite this reference in revision manuscript .
8. Please include a discussion why Kd measurements are not included in this study? and justification why they are not necessary
Response: Reviewer’s comment is well, Kd measurements was very impartment for binding affinity in ligand and receptor. We also tried to detect the affinity between Venetoclax and spike protein, unfortunately, the purification and analysis technology of the protein is still in the testing stage. Therefore, we use molecular docking to calculate binding affinities and ELISA to analyze binding free energy. The data showed that Venetoclax can inhibit the binding between Spike protein and ACE2.
9. Please, reformulate the conclusions subsection to highlight the most important points achieved by your analyses. Quantitative values must support your conclusions.
Response: Following the Reviewer’s comment, we reformulated new conclusions subsection in revised manuscript.

Reviewer 4 Report
Although the subject question - a computational search in databases for drugs that inhibit Spike-ACE2 interaction with following biochemical - cell biological examination of top ranking compound is relevant, but performed cell biology experiments are not convincing can be said irrelevant in respect of specific interaction between venetoclax and spike protein. Unfortunately only venetoclax had been analysed in detail but Venetoclax has been described earlier for this purpose (PMID: 32221306 ), therefore the novelty of this paper is restricted.
- In computational work some informations are missing.
- Analysis of the assay results are not convincing, statements are not relevant in several points, should be rephrased. First of all Venetoclax had only a very slight inhibitory effect on Spike expression - but investigating the expression of a totally foreign protein in an overexpressing cell is not very informative in this way, may be venetoclax can effect any kind of overexpressed protein. A kind of control is required in this respect.
- Overexpressed Spike contain a Flag-tag, but there is no explanation of 2 labeled lines on the western by anti-Flag antibody. It shoud be explained.
- For some reason Mg132 (proteasome inhibitor) treatment inhibited the presence of the full length form - it seems to be independent of venetoclax, contrary to the explanations phrased in the paper.
- Details of venetoclax treatment are not given precisely ( - how long was it used).
- The applied ELISA test for investigation of binding is specified on its homepage as a qualitative test, and it is not sure that it is applicable for drug testing at all. Instead of the presented timeline of this assay (with huge time frames) a concentration dependence curve would be more convincing.
- The paper does not give any statistical evidence, one presented western or IF picture is not convincing.
- Some of the pictures are of bad quality - Figure 3 shows a heavily overstained blots. Based on this figure 3. not much can be deduced, probably mutant versions of spike are not expressed well and full length form is missing with mutants. Again specific interaction of venetoclax and mutant spike proteins can not be analysed by this experiments.
- The paper's english have to be significantly improved.
Author Response
Reviewer 4.
Although the subject question - a computational search in databases for drugs that inhibit Spike-ACE2 interaction with following biochemical - cell biological examination of top ranking compound is relevant, but performed cell biology experiments are not convincing can be said irrelevant in respect of specific interaction between venetoclax and spike protein. Unfortunately only venetoclax had been analysed in detail but Venetoclax has been described earlier for this purpose (PMID: 32221306 ), therefore the novelty of this paper is restricted.
Response: Following the Reviewer’s comment, Venetoclax (ABT-199、GDC-0199、RG7601) is currently the only approved BCL-2 inhibitor that can cause tumor cells to produce apoptosis and achieve the effect of killing tumors. However (PMID: 32221306 ) this paper is mainly focus on Broad-spectrum cysteine protease inhibitor E64D (#HY-100229), Cathepsin L-specific inhibitor SID 26681509 (#HY-103353), Cathepsin B-specific inhibitor CA-074 (#HY-103350), PIKfyve inhibitors apilimod (#HY-14644) and YM201636 (#HY-13228), Calcium channel blocker tetrandrine (#S2403) and does not mention Venetoclax. Therefore, we believe that Venetoclax's research on SARS-CoV-2 is novel.
1. In computational work some informations are missing.
Response: Following the Reviewer’s comment, we have added the new computational methods in the revised manuscript.
2. Analysis of the assay results are not convincing, statements are not relevant in several points, should be rephrased. First of all Venetoclax had only a very slight inhibitory effect on Spike expression - but investigating the expression of a totally foreign protein in an overexpressing cell is not very informative in this way, may be venetoclax can effect any kind of overexpressed protein. A kind of control is required in this respect.
Response: Following the Reviewer’s comment, western blotting band was quantified by NIH software (ImageJ) We have added the reduction ratio to the manuscript. In addition, our study found that the protein expression of ACE2 was not effected by Venetoclax in the HEK293T cells (data not show). In this study, we use virtual drug screening to establish pharmacophore groups and analyze the ACE2 binding site of Spike protein by molecular chimerization and molecular dynamic simulation. Screening results found that Venetoclax has the potential binding ability to Spike protein of SARS-CoV-2. We also analyzed binding assay between SARS-CoV-2 spike-protein and ACE2 receptor. The results found the Venetoclax decreased the binding in between SARS-CoV-2 spike-protein and ACE2 receptor. Therefore, Our results suggest that Venetoclax as a candidate for clinical prevention and treatment.
3. Overexpressed Spike contain a Flag-tag, but there is no explanation of 2 labeled lines on the western by anti-Flag antibody. It shoud be explained.
Response: Following the Reviewer’s comment, we added” Anti-Flag-M2 antibody detected two major bands, 180 kDa and 90 kDa, reflecting full-length and cleaved Spike protein, respectively.” in the materials and methods section.
4. For some reason Mg132 (proteasome inhibitor) treatment inhibited the presence of the full length form - it seems to be independent of venetoclax, contrary to the explanations phrased in the paper.
Response: Following the Reviewer’s comment, we modify the section “Previous studies have found that Venetoclax in combination with proteasome inhibitors can improve the effect of treatment35. Therefore, we believe that ubiquitin ligase can label endogenous proteins and degrade them. The results data found that proteasome inhibitor MG132 increased the degradation of Spike protein when Venetoclax treatment the cell (Figure 5E)”
5. Details of venetoclax treatment are not given precisely ( - how long was it used).
Response: Following the Reviewer’s comment, we simulate the administration of Venetoclax pre-exposure (prophylactic) and post-exposure (treatment) the infection of the COVID-19 virus and detect the expression level of Spike protein in the cell (Figure 5C).
6. The applied ELISA test for investigation of binding is specified on its homepage as a qualitative test, and it is not sure that it is applicable for drug testing at all. Instead of the presented timeline of this assay (with huge time frames) a concentration dependence curve would be more convincing.
Response: Following the Reviewer’s comment, we use in vitro experiments to verify the important role of Venetoclax in the cell. We use HEK293T cells for drug killing and gene transfer experiments. First, we using CCK-8 to analyze whether Venetoclax has the ability to inhibit the growth of 293T cells. The results found that dose-dependent (5, 10, 20 and 40 nM) of Venetoclax will not affect the growth of 293T cells (Figure 5A). We found that 5, 10, 20 and 40 nM was a safe concentration that is harmless to the human body, and then validation the drug effect with huge time frames.
7. The paper does not give any statistical evidence, one presented western or IF picture is not convincing.
Response: Following the Reviewer’s comment, western blotting band and immunofluorescence intensity was quantified by NIH software (ImageJ).
8. Some of the pictures are of bad quality - Figure 3 shows a heavily overstained blots. Based on this figure 3. not much can be deduced, probably mutant versions of spike are not expressed well and full length form is missing with mutants. Again specific interaction of venetoclax and mutant spike proteins can not be analysed by this experiments.
Response: Following the Reviewer’s comment, the Full-length gels and blots was showed in the supplemental data. In addition, the Reviewer’s comment is well, mutant versions of spike are not expressed well. Therefore, we use molecular simulations to assist the results. We adopted the BAR protocol to calculate the binding affinity differences between S1-RBD and Venetoclax and the point mutations. The resulting binding free energy differences (ΔΔG) between WT- and Q493Y-, S494R- and Q493Y/S494R-MT were 3.34 ± 4.06, -7.86 ± 3.96 and -3.27 ± 5.12 kJ/mol, respectively (Figure 6B, C and D). The Venetoclax binding affinity was significantly increased in the S494R-MT (ΔΔG < 0). In addition, western blotting results(180 kDa) also consistent with our binding free energy experiments and found that S493R has a higher effect to decrease the expression of Spike protein compare to Q493Y and Q493Y/S493R (Figure 6E, F, G and H) in prophylactic and treatment group. Above results we believe that SARS CoV-2-S493R increase the activity of Venetoclax drug resulting in decreased the protein level of SARS-CoV-2 spike-protein.
9. The paper's english have to be significantly improved.
Response: Following the Reviewer’s comment, the manuscript has now been professionally edited with the grammar checked.
Reviewer 5 Report
Chen et al performed some in silico docking based on “so claimed binder”(which were not established), and claimed Venetoclax “degradation” the expression of spike-protein through amino acids Q493 and S494. This title is not clear to me at all. The in vitro experiments are very rough. No convincingly data support that the Venetoclax can really bind to spike or ACE2.
Author Response
Reviewer 5.
Chen et al performed some in silico docking based on “so claimed binder”(which were not established), and claimed Venetoclax “degradation” the expression of spike-protein through amino acids Q493 and S494. This title is not clear to me at all. The in vitro experiments are very rough. No convincingly data support that the Venetoclax can really bind to spike or ACE2.
Response: Following the Reviewer’s comment, we revision the title “Venetoclax decrease the expression of spike-protein through amino acids Q493 and S494 in SARS-CoV-2”. In this study, we use virtual drug screening to establish pharmacophore groups and analyze the ACE2 binding site of Spike protein by molecular chimerization and molecular dynamic simulation. Screening results found that Venetoclax has the potential binding ability to Spike protein of SARS-CoV-2. We also analyzed binding assay between SARS-CoV-2 spike-protein and ACE2 receptor. The results found the Venetoclax decreased the binding in between SARS-CoV-2 spike-protein and ACE2 receptor.
Round 2
Reviewer 2 Report
All my comments were resolved and I have no further concerns.
Author Response
All my comments were resolved and I have no further concerns.
Response: Thank the reviewers’ comments.
Reviewer 4 Report
At first I would like to admit, that there is a kind of novelty in studying the effect of venetoclax in covid, I was not precise enough, moreover I gave a wrong link. I read about venetoclax in relation to covid in the following paper: https://journals.asm.org/doi/full/10.1128/mBio.03681-20.
In this paper it is stated „Antiviral potency of drug hits against SARS-CoV-2 infection of Vero-E6 cells.Of the 22 compounds identified through in silico and SPR screening, 11 that showed blocking in SPR studies and that were not overtly toxic and did not have solubility issues were tested against SARS-CoV-2 infection of Vero-E6 cells (Fig. 4).Out of the 11 compounds tested, 4 were found to be toxic at the highest concentration tested while inactive at lower concentrations—velpatasvir, simeprevir, acalabrutinib, and venetoclax.”
After molecular docking studies they investigated SPR competition study using immobilized ACE2 that is a more suitable assay for such purposes than ELISA used in the present paper. They also used Vero-6 cell studies for identification of inhibiton of viral infection that is missing from the present paper.
This paper is the closest in subject, but some others also investigate venetoclax related to covid as presented below:
https://www.science.org/doi/full/10.1126/sciadv.abf8609
When we monitored cell death in Vero cells following SARS-CoV-2 infection, we found that treatment with ABT-199 (an inhibitor of BCL-2) or ABT-737 (an inhibitor of BCL-2, BCL-XL, and BCL-w) substantially enhanced virus-induced cell death relative to vehicle control
https://www.sciencedirect.com/science/article/pii/S0166354221000450
ABT-199, also known as venetoclax, is a potent selective Bcl2 inhibitor, which induces the apoptosis pathway. An early work showed that Bcl2 expression prevents SARS-CoV1 induced apoptosis (Bordi et al., 2006). In addition, previous reports demonstrated that SARS-CoV1 7a protein was dependent on Bcl2 to induce apoptosis, suggesting Bcl2 as an important host factor for virus replication and pathogenesis (Tan et al., 2007). However, despite its good EC50 (about 6.2 μM) against SARS-CoV2, venetoclax shows high toxicity in the tested cells.
https://journals.plos.org/Plospathogens/article?id=10.1371/journal.ppat.1009898
We next investigated antiviral activity of selected drugs in Vero E6 cells.Strong inhibition was detected for ethacridine with 5–6 logs reduction in viral titer, simeprevir ~4-log reduction, ABT-199 ~2-log reduction, hydroxyprogesterone ~1-log reduction, cinacalcet ~1-log reduction.
Four of them, including simeprevir, ABT-199, hydroxyprogesterone and cinacalcet effectively inhibited Mpro and blocked SARS-CoV-2.
https://journals.asm.org/doi/full/10.1128/mBio.03681-20
Antiviral potency of drug hits against SARS-CoV-2 infection of Vero-E6 cells.Of the 22 compounds identified through in silico and SPR screening, 11 that showed blocking in SPR studies and that were not overtly toxic and did not have solubility issues were tested against SARS-CoV-2 infection of Vero-E6 cells (Fig. 4).Out of the 11 compounds tested, 4 were found to be toxic at the highest concentration tested while inactive at lower concentrations—velpatasvir, simeprevir, acalabrutinib, and venetoclax.
https://www.ncbi.nlm.nih.gov/pmc/articles/PMC7430593/
other agents identified in our screen including cilnidipine, dasatinib and venetoclax were also effective in reducing viral entry (fig. S7c, d).
Nucleocapsid protein has also been extensively studied for screening out the effective therapeutic options for the ongoing pandemic. About 24 unlike drug molecules have been reported targeting nucleocapsid protein. Venetoclax is the most frequently reported drug substrate for this protein suggested by computational high throughput screening.
https://www.ncbi.nlm.nih.gov/pmc/articles/PMC7062204/
The results of virtual screening of drugs on the active sites of SARS-CoV-2 3CL pro model. In Table 3 venetoclax is mentioned among the result of screening
Regarding to answers of other points
- I accept changes in the description of computation part, it is much better now (although I am not an expert of this field)
- Regarding the experimental part I still have serious critics. I accept that the authors used software to investigate band strength, but westerns should be repeated with similar results at least by 3 times and it is not stated anywhere that how many times was it repeated. Moreover the whole assay design could be better. In the study, Spike protein is overexpressed in a model cell from an artificial strong promoter and for the sake of some kind of similarity to a patient situation they applied venetoclax long time before or after transfection, but transfection with a spike containing vector is no similarity to a viral infection. Usually 4-6 hours long drug treatments are applied after transfection in a standard experiment. Pretreatment of cells by drug before the transfection is unusual and the fact that venetoclax reduced the amount of detected Spike level does not mean any direct interaction. The proper control would be to investigate the level of another similar size and localized protein expressed from the same vector. In Fig 5D – although it is not clear but as I understand 0-4 ug means the quantity of transfected DNA – it is strange that at 0 ug (no vector ctr) the lower band still can be seen – that means this is a non-specific band, not Spike! Only the upper band can be the full length Spike. The ELISA assay used in this paper (shown in Fig 5B) is not a quantitative assay based on internet information therefore probably not suitable to test small molecular inhibitors for ACE2-Spike interaction, anyway the overall 5-10% inhibition is very week if it means anything at all, there is no control for this experiment and only 2 small concentrations were investigated, instead of a 4-5 points concentration curve. Unacceptable based on scientific standards. Fig 5E – for some reason Mg132 treatment (only before transfection!) inhibited Spike expression – the reason is not clear, but presumably MG132 reduced the transfection efficiency (may be it effects endocytosis) and has nothing to do with Spike degradation by proteasomes. Protein degradation effect should be present when a transfected cell is treated by proteasome inhibitor, but careful design is needed when overexpressed protein is investigated to see the effect. Fig 5F - Based on earlier results – if venetoclax decreased the level of Spike protein it is trivial that less double staining is resulted that does not suggest a decreased interaction, although the ratio of proteins can be important in such studies. Statistical evaluation of costaining is missing.
- The 2 bands could be the full length and cleaved Spike based on mol weight, but as I wrote earlier even non transfected cell contains the lower band. Fig 6, westerns are heavily overstained, numerous unspecific bands can be seen. At 180 kDa band it can be seen that only wt Spike is expressed well, the mutants have a very faint expression. So regarding the mutants the Spike folding seems to be the main problem, not receptor binding, that can be problematic as well of course but it comes later. Here I agree with writers that computational modeling is in harmony with experimental results.
- English language is still not enough good, just an example at row 274-275. Several letter faults are also present for example instead of venetoclax ventoclax was used in some figures. In Fig 5 instead of inhibiton inhibitation was wrote.
Sorry to say, but experimental part of the presented paper is still unacceptable because of lacking proper controls and statistical analysis. The presented experiments are not suitable to present specific and strong interation between spike protein and venetoclax deduced from computational score values.
